# Associations of Tumor Necrosis Factor-Alpha Gene Polymorphisms *(TNF)-α* TNF-863A/C (rs1800630), TNF-308A/G (rs1800629), TNF-238A/G (rs361525), and TNF-Alpha Serum Concentration with Age-Related Macular Degeneration

**DOI:** 10.3390/life12070928

**Published:** 2022-06-21

**Authors:** Guoda Zazeckyte, Greta Gedvilaite, Alvita Vilkeviciute, Loresa Kriauciuniene, Vilma Jurate Balciuniene, Ruta Mockute, Rasa Liutkeviciene

**Affiliations:** 1Medical Academy, Lithuanian University of Health Sciences, Eiveniu Str. 2, LT-50161 Kaunas, Lithuania; gouda.zazeckyte@stud.lsmuni.lt (G.Z.); ruta.mockute@gmail.com (R.M.); 2Neuroscience Institute, Medical Academy, Lithuanian University of Health Sciences, Eiveniu Str. 2, LT-50161 Kaunas, Lithuania; alvita.vilkeviciute@lsmuni.lt (A.V.); loresa.kriauciuniene@lsmuni.lt (L.K.); rasa.liutkeviciene@lsmuni.lt (R.L.); 3Department of Ophthalmology, Medical Academy, Lithuanian University of Health Sciences, Eiveniu Str. 2, LT-50161 Kaunas, Lithuania; jurate.balciuniene@lsmuni.lt

**Keywords:** age-related macular degeneration (AMD)-1, *TNF*-*α*-2, rs18006303-3, rs1800629-4, rs361525-5

## Abstract

Age-related macular degeneration (AMD) is a neurodegenerative disease leading to irreversible central vision loss among the elderly in developed countries. While the disease accounts for 9% of all cases of vision loss, the prevalence of AMD is likely to increase due to the exponential aging of the population. Due to this reason, our study aimed to determine the associations of tumor necrosis factor-alpha (*TNF-α*) gene single-nucleotide polymorphisms (SNPs) TNF-863A/C (rs1800630), TNF-308A/G (rs1800629), TNF-238A/G (rs361525), and TNF-α serum concentration with age-related macular degeneration. Analysis of *TNF-α* rs1800630, rs1800629, and rs361525 polymorphisms showed that the *TNF-α* rs1800630 A allele was statistically significantly more frequent in the exudative AMD group compared to the control group (*p* = 0.029). Additionally, the *TNF-α* rs1800630 A allele was more frequent in females with exudative AMD than in the control group of healthy females (*p* = 0.027). The *TNF-α* rs1800630 A allele was more frequent in females with exudative AMD than in females with early AMD (*p* = 0.014). *TNF-α* rs1800630, rs1800629, and rs361525 haplotype A-A-G were associated with decreased odds of exudative AMD (*p* < 0.0001), and haplotype A-G-G was associated with 24-fold increased exudative AMD occurrence (*p* < 0.0001). TNF-α protein levels were lower in subjects with exudative AMD compared to the control group (*p* < 0.001). The study showed significant associations between inflammatory cytokine *TNF-α* single-nucleotide polymorphisms and serum level with AMD pathogenesis. Analysis of *TNF-α* genotypes and serum concentration may be helpful for the AMD diagnosis.

## 1. Introduction

Aging-related disorders can be defined as the progressive occurrence of several defective cellular mechanisms or metabolic pathways, resulting in degeneration [1]. Age-related macular degeneration (AMD) is a neurodegenerative disease that is the leading cause of irreversible central vision loss among the elderly in developed countries [2,3]. AMD is the third leading cause of blindness globally, following cataracts and glaucoma [4]. The disease accounts for some 9% of all cases of vision loss. In 2020, approximately 200 million people were affected by AMD worldwide [5]. The prevalence of AMD is likely to increase due to the exponential aging of the population [6]. AMD’s most common characteristic features can be recognized in drusen and the growth of choroidal vessels (choroidal neovascularization) [7]. Drusen is an active and inactive complement associated with inflammatory products, an aggregate of lipoprotein, cell debris, oxysterols, oxidized phospholipids, and Alu RNA deposits, which begin to emerge later in life and not during development [1]. These aggregates deposit between the basement membrane of the retinal pigment epithelium (RPE) and the inner collagen layer of Bruch’s membrane beneath the RPE [8]. The presence of drusen within the macula is the hallmark sign of AMD [9]. However, the existence of a few small hard drusen in the peripheral retina is considered a normal part of the aging process. Nevertheless, large and many drusen in the macula signify early AMD [10].

Various pathologies, including a focal detachment of the RPE, outer retinal atrophy, and new blood vessel growth between Bruch’s membrane and the retina, can progress into either geographic atrophy (GA) or choroidal neovascularization (CNV) AMD, which are also known as ‘dry’ or ‘wet’ AMD, respectively [9]. The primary clinical characteristic of late-stage ‘dry’ AMD is GA. It is characterized by oval areas of hypopigmentation and is usually the consequence of RPE cell loss. CNV is the defining characteristic of late-stage ‘wet’ or neovascular AMD. The neovascularization has two etiologic patterns: (1) new vessels sprouting from the choroidal vessels, penetrating Bruch’s membrane, and growing into the subretinal space are the classical descriptions of wet AMD, which is most common; and (2) vessels that are derived mainly from the retinal circulation in a process that has been called retinal angiomatous proliferation (RAP) [10].

AMD is a multifactorial disease caused by various genetic variants. Each has a modest effect on the risk and is also influenced by nongenetic/environmental factors (aging, smoking, family history, hypertension, etc.) [7,11]. The genetic component of AMD has been estimated at 45% to 70% [12]. Twin studies demonstrating greater concordance in monozygotic (37%) than dizygotic (19%) twins, and studies showing clustering of AMD in families, a hallmark of a disease with complex inheritance, were the first to underscore the genetic basis for AMD [13]. It has been shown that an individual with a sibling or a parent with AMD is 12–27 times more susceptible than someone from the general population to develop AMD [14]. As of this writing, 34 genetic loci, encompassing 52 gene variants, have been associated with AMD; it has been estimated that these 52 variants collectively account for about half of the heritability of the disease. These genes and genomic regions may be divided into high-effect, low-effect, and unknown variants. The two most widely studied and important loci, due to their large effect sizes and relatively high frequencies in the population, are complement factor H (*CFH*) and age-related maculopathy susceptibility 2 (*ARMS2*) [12]. However, the variation in such genomic regions alone cannot predict disease development with high accuracy. Therefore, current genetic studies aim to identify new genes associated with AMD and their modifiers to discover diagnostic or prognostic biomarkers [15].

Recent studies have shown the immune system’s role in AMD development and progression. Increased concentrations of several inflammatory cytokines have been found both in serum and locally in ocular tissues or fluids in patients with AMD [16]. An example is tumor necrosis factor-alpha (TNF-α). It belongs to the group of proinflammatory cytokines and appears to participate in the pathogenesis of inflammatory, edematous, neovascular, and neurodegenerative diseases [17]. TNF-α is a pleiotropic cytokine produced by many different cells in the body [18]. However, the major producers are macrophages, monocytes, neutrophils, T cells, and NK cells [19]. Transcription of the *TNF-α* gene is genetically regulated, and polymorphisms in the promoter region may alter TNF-α production [16].

Due to the vital role of inflammatory molecules in the pathogenesis of AMD, we aimed to determine the associations of tumor necrosis factor-alpha (*TNF-α*) gene single-nucleotide polymorphisms (SNPs) TNF-863A/C (rs1800630), TNF-308A/G (rs1800629), TNF-238A/G (rs361525), and TNF-α serum concentration with AMD.

## 2. Materials and Methods

### 2.1. Study Subjects

The Ethics Committee approved the study for Biomedical Research, Lithuanian University of Health Sciences (No. BE-2-/48).

The study included subjects admitted to the Hospital of Lithuanian University of Health Sciences Ophthalmology Department for preventive ophthalmological evaluation. In total, 1078 participants were enrolled in our study. Polymorphisms were determined by dividing the subjects into three groups. The first group consisted of patients with early AMD (n = 330) ranging in age from 42 to 94 years. This group included 227 (68.8%) females and 103 (31.2%) males. The second group consisted of patients with exudative AMD (n = 393) ranging in age from 49 to 95 years. The group included 254 (64.6%) females and 139 (35.4%) males. Moreover, the third group consisted of ophthalmologically healthy individuals (n = 355) aged 51 to 94 years. The group included 222 (62.5%) females and 133 (37.5%) males. TNF-α serum concentration was determined in subjects divided into two groups: subjects with exudative AMD (n = 18) and ophthalmologically healthy individuals corresponding to the subjects by age and gender (n = 20).

The AMD group consisted of subjects who underwent ophthalmological evaluation and were diagnosed with early or exudative AMD. Present study subjects were evaluated by slit-lamp biomicroscopy. Additionally, all AMD patients underwent optical coherence tomography (OCT), and optical coherence tomography angiography (OCT-A) was performed to confirm the exudative AMD after the OCT examination.

The Age-Related Eye Disease Study (AREDS) classification system was used for AMD diagnosis and classification [20].

Early AMD consists of a combination of multiple small drusen (protein and lipid deposit) formation between the RPE and BrM [21] and several intermediate (63–124 μm in diameter) drusen or retinal pigment epithelial abnormalities; may not cause any symptoms.The intermediate form is described as a presence of at least one large (≥125 μm in diameter) drusen, numerous medium-sized drusen, or GA without extension to the center of the macula with mild symptoms, as mild blurriness in their central vision or trouble seeing in low lighting or may not cause any symptoms.The advanced AMD is divided into:
Dry/atrophic AMD with the GA of the RPE;Neovascular or exudative AMD, which is diagnosed when choroidal neovascularization with detachments in the RPE hemorrhages and/or scars appear and cause progressive blurring or other central vision impairments [22].


AMD exclusion criteria

Unrelated eye disorders, e.g., high refractive error, cloudy cornea, lens opacity (nuclear, cortical, or posterior subcapsular cataract) except minor opacities, keratitis, acute or chronic uveitis, glaucoma, or diseases of the optic nerve.Any other inflammatory diseases.Systemic illnesses, e.g., diabetes mellitus, malignant tumors, systemic connective tissue disorders, chronic infectious and noninfectious diseases, hypertension, coronary artery disease, stroke or conditions following organ or tissue transplantation.Ungraded color fundus photographs resulting from obscuring the ocular optic system or because of fundus photograph quality.Use of antiepileptic or sedative drugs.

### 2.2. Deoxyribonucleic Acid Extraction

The salting-out method performed the extraction of deoxyribonucleic acid (DNA). Blood was collected in vacuum tubes with the anticoagulant EDTA (ethylenediaminetetraacetate) to prevent microspheres’ formation and protect the DNA from degradation. The DNA used in the study was isolated from peripheral venous white blood cells. The salting-out method is based on collecting cells by centrifugation, their suspension in a buffer solution, the degradation of cell membranes with detergents, the hydrolysis of proteins by proteinase K, the deproteinization of chloroform, and the precipitation of DNA with ethanol. It is known that about 250 μg of DNA can be obtained from 10 mL of blood and is used in assays such as real-time polymerase chain reaction (RT-PCR). In the first stages of DNA extraction, it is vital to inactivate nucleases (enzymes that destroy DNA or RNA) using different buffers. It is also recommended that all DNA extraction steps be performed at a low temperature, and extracted DNA be stored at −70 °C and unfroze before use.

### 2.3. Measurement of DNA Concentration by Spectrophotometer

For further research using DNA, it is necessary to measure the concentration of extracted DNA. It was measured using an Agilent Technologies Cary 60 UV-Vis spectrophotometer, which can analyze minimal amounts of samples (1–2 µL) without using cuvettes or capillaries. By measuring the absorbance (optical density) of the solution at a certain ultraviolet (UV) wavelength, the DNA purity and the amount of remaining protein were determined along with the DNA concentration. 260 nm UV light waves are absorbed by nucleic acids (DNA and RNA) and 280 nm by proteins. The ratio between DNA and protein absorption (260/280) should be 1.8.

### 2.4. Detection of Single-Nucleotide Polymorphisms by a RT-PCR

TNF-α gene SNPs: TNF-863A/C (rs1800630), TNF-308A/G (rs1800629), TNF-238A/G (rs361525) were studied by RT-PCR. A series of three steps achieve RT-PCR amplifications: (1) denaturation, in which double-stranded DNA templates are heated to separate the strands; (2) annealing, in which short DNA molecules called primers bind to flanking regions of the target DNA; and (3) elongation, in which DNA polymerase extends the 3′ end of each primer along the template strands. These steps are repeated to produce exact copies of the target DNA exponentially. Samples of 1078 subjects were genotyped with the StepOnePlus RT-PCR amplifier (Applied Biosystems by Thermo Fisher Scientific, Singapore). Applied Biosystems and Thermo Fisher Scientific (Waltham, MA, USA) developed primers and molecular markers for genotyping.

### 2.5. Determination of Serum TNF-α Protein by ELISA

ELISA is an enzyme-linked immunosorbent assay for detecting and quantifying peptides, secreted proteins, hormones, and cytokines. The most crucial element of an ELISA is a particular antibody–antigen interaction. A commercial TNF-alpha Human ELISA Kit (Catalog No.: BMS223-4; Thermo Fisher Scientific (Vienna, Austria)) was used to measure serum TNF-α levels in subjects. TNFα solid-phase sandwich ELISA is designed to measure the amount of the target bound between a matched antibody pair. A target-specific antibody has been precoated in the wells of the microplate. Samples and standards were then added into wells and bound to the immobilized antibody. The sandwich was formed by adding the second (detector) antibody. The enzyme activity was measured using a substrate that changes color when modified by the enzyme.

### 2.6. Statistical Analysis

Statistical analysis was performed using the IBM SPSS Statistics 27.0 software. Data are presented as absolute numbers with percentages. Descriptive statistical characteristics were used for non-normally distributed data: median, minimum, and maximum (min. and max.) values. Mann–Whitney U test was used to determine the differences in age and TNF-α concentration between two independent groups. The control group’s distribution of TNF-α rs1800630, rs1800629, and rs361525 polymorphisms were evaluated using the Hardy–Weinberg equilibrium (http://www.oege.org/software/hwe-mr-calc.shtml accessed on 11 March 2022). The Pearson χ^2^ test using two-way alternatives compared the distribution of polymorphism genotypes and alleles between patients with early and exudative AMD and the control group. Based on genetic models, a binary logistic regression analysis was performed to assess AMD’s odds ratios (OR). This analysis was performed for individual AMD groups (early and exudative), indicating OR with a 95% confidence interval (CI). Logistic regression analysis for exudative AMD was performed with age adjustment, as subjects in the exudative AMD group were statistically significantly older than the control group. The best genetic model selection was based on the Akaike Information Criterion (AIC). Therefore, the best genetic models were those with the lowest AIC values. Haplotype analysis was performed using the online program SNPStats (https://www.snpstats.net/start.htm accessed on 1 February 2022). Linkage disequilibrium (LD) analysis was assessed by D′ (deviation between the expected haplotype frequency and the observed frequency) and r^2^ (square of the haplotype frequency correlation coefficient) measures. Associations of haplotypes with early and exudative AMD were assessed by logistic regression, indicating OR with a 95% CI. Statistically significant differences were observed when *p* < 0.05.

## 3. Results

Our study enrolled 1078 subjects, who were divided into three groups: the control group (n = 355), patients with early AMD (n = 330), and patients with exudative AMD (n = 393). The demographics are shown in Table 1. There were no statistically significant differences between females and males in the control and early AMD and control and exudative AMD groups (*p* = 0.085; *p* = 0.552). Additionally, any differences were found comparing the age of the control and early AMD groups (*p* = 0.266). However, the patients with exudative AMD were statistically significantly older than controls (*p* < 0.001). Thus, further exudative AMD analyses were adjusted by age.

We evaluated the distributions of *TNF-α* rs1800630, rs1800629, and rs361525 genotypes in the control group using the Hardy–Weinberg equilibrium (HWE). The analysis showed that the SNPs were in HWE (*p* > 0.01) (Appendix A).

Analysis of *TNF-α* rs1800630, rs1800629, and rs361525 revealed that the *TNF-α* rs1800630 A allele was statistically significantly more frequent in the exudative AMD group compared to the control group (19.3% vs. 15.1%, *p* = 0.029) (Table 2). No statistically significant differences were found between the genotypes and alleles distribution of *TNF-α* rs1800630, rs1800629, and rs361525 when comparing early AMD and control groups, as well as comparing exudative AMD and control groups. The distribution did not statistically significantly differ between early AMD and exudative AMD groups (Appendix A).

Binary logistic regression of *TNF-α* rs1800630, rs1800629, and rs361525 polymorphisms in the early AMD and control groups and the exudative AMD and control groups did not show statistically significant results (Appendix A).

We performed the haplotype analysis of TNF-863A/C (rs1800630), TNF-308A/G (rs1800629) and TNF-238A/G (rs361525) polymorphisms. The deviation between the predicted haplotype frequency and the observed frequency (D′) was calculated, and the square of the correlation coefficient (r^2^) was estimated. Data are presented in Table 3 and Table 4.

We analyzed haplotype frequencies, and the results did not reveal any differences between patients with early AMD and the control group (Appendix A). On the other hand, statistical analysis of exudative AMD have shown that haplotype A-A-G of *TNF-α* (rs1800630, rs1800629, and rs361525) is associated with the decreased odds of exudative AMD development (OR = 0.12; 95% CI: 0.05–1,29; *p* < 0.0001), and the haplotype A-G-G of *TNF-α* (rs1800630, rs1800629, and rs361525) is associated with increased odds of exudative AMD development (OR = 24.45; 95% CI: 9.39–63.63; *p* < 0.0001) (Table 5).

Continuing the analysis of the study, we evaluated the associations of single-nucleotide polymorphisms (*TNF-α* rs1800630, rs1800629, rs361525) with the predisposition to early and exudative AMD occurrence in males and females separately.

We compared the genotypes’ and alleles’ frequency distributions of the *TNF-α* rs1800630, rs1800629, and rs361525 polymorphisms in the early and exudative AMD and control groups by gender. SNPs analysis did not reveal statistically significant results when comparing early AMD and control groups according to gender (*p* > 0.05) (Appendix A).

No statistically significant associations were found between male and female genotypes’ frequency analysis comparing controls and exudative AMD group (*p* > 0.05) (Table 6). Analysis revealed that the *TNF-α* rs1800630 A allele was more frequent in females with exudative AMD than in the control group (20.9% vs. 15.3%, *p* = 0.027) (Table 6).

No statistically significant associations in genotype frequency distribution were found between males and females with early AMD and exudative AMD (*p* > 0.05) (Table 7). Allele frequency analysis showed that the *TNF-α* rs1800630 A allele was statistically more frequent in females with exudative AMD than in early AMD (20.9% vs. 14.8%, *p* = 0.014) (Table 7).

We performed the binary logistic regression analysis to evaluate these SNPs’ impact on early and exudative AMD by gender. The analysis did not show associations between *TNF-α* rs1800630, rs1800629, and rs361525 polymorphisms and the early or exudative AMD (Appendix A).

TNF-α protein serum levels were measured in patients with exudative AMD who had not been treated with anti-VEGF injections (n = 18) and controls (n = 20). We found a statistically significant difference between study groups: TNF-α serum levels were lower in patients with exudative AMD than in the control group (mean ± standard deviation (SD): 16.182 pg/mL ± 6.094 vs. 30.652 pg/mL ± 6.322; *p* < 0.001). The results are shown in Figure 1.

We also compared the TNF-α serum levels for all patients (exudative AMD and control groups) and only for exudative AMD patients and only for the control group, according to the genotypes of *TNF-α* rs1800630, rs1800629, and rs361525. Due to the small sample size, we formed two groups for comparison: heterozygotes and homozygotes according to a rarer allele and homozygotes according to a more frequent allele. No statistically significant differences were found (Appendix A).

## 4. Discussion

Recent studies of the Russian population have shown that promoter polymorphisms at -238 (rs361525), -308 (rs1800629), and -863 (rs1800630) positions of its gene could regulate TNF-α production. These genetic variants may have implications for AMD pathogenesis due to inflammatory processes imbalance caused by TNF-α production dysregulation [16]. It has been reported that the genetic alterations in the *TNF-α* locus are involved in high TNF-α production [19]. In this study, we aimed to determine the association between TNF-863A/C (rs1800630), TNF-308A/G (rs1800629), and TNF-238A/G (rs361525) single-nucleotide polymorphisms, TNF-α serum levels and the pathogenesis of age-related macular degeneration. Based on previous research, *TNF-α* rs3615225 has been associated with gastric cancer and sepsis risk, particularly in eastern populations [23,24]. *TNF-α* rs1800629 polymorphism was significantly associated with the risk of chronic periodontitis, type 2 diabetes mellitus, and celiac disease [25,26]. The analysis of Korean females showed that the *TNF-α* rs1800630 polymorphism was significantly related to the increased birth weight (g) within preterm-birth (PTB) patients [27]. Moreover, rs1800630 polymorphism is linked to an elevated ankylosing spondylitis susceptibility in Asians [28].

Analysis of *TNF-α* rs1800630, rs1800629, and rs361525 polymorphisms showed that the *TNF-α* rs1800630 A allele was statistically significantly more frequent in the exudative AMD group than in the control group (19.3% vs. 15.1%, *p* = 0.029). Additionally, the *TNF-α* rs1800630 A allele was statistically significantly more frequent in females with exudative AMD than in the control group of healthy females (20.9% vs. 15.3%, *p* = 0.027). Moreover, we found statistically significant differences in the *TNF-α* rs1800630 polymorphism allele frequencies between females with primary AMD and females with exudative AMD. The A allele is more frequent in females with exudative AMD compared to females with early AMD (20.9% vs. 14.8%, *p* = 0.014).

Studies show that the frequency distribution of allele/genotype varies among different ethnic and racial groups. The prevalence of the TNF–308 A allele was 7% in China and Korea, 15% in Germany, and 11% in normal Italian subjects. The Portuguese, Finnish, and US populations have 13% and 14%, and 15% of TNF–308 A allele frequencies. In India, the –308 A allele frequency falls between the values for China/Korea and Italy [29]. The group of researchers reported that among six candidates for TNF-α gene SNP genetic markers (−238 G/A, −308 G/A, +489 G/A, −857 C/T, −863 C/A, and −1031 T/C), only −1031 T/C was significantly associated with the occurrence of wet AMD in the population of Taiwan Chinese [16]. Other studies show that the −1031 T/C polymorphism of the TNF-α gene does not play an essential role in developing dry AMD in the population of northeastern Iran [16]. This phenomenon can be explained not only by ethnic differences within groups but also by differences in sample size, methodological differences, and dominance of different etiological factors between populations.

The haplotype frequencies of TNF-863A/C (rs1800630), TNF-308A/G (rs1800629), and TNF-238A/G (rs361525) SNPs were not involved in previous AMD studies. Our results revealed that the haplotype A-A-G of TNF-863A/C (rs1800630), TNF-308A/G (rs1800629), and TNF-238A/G (rs361525) polymorphisms was associated with the decreased odds of exudative AMD (OR = 0.12, *p* < 0.0001), and haplotype A-G-G with the increased odds of exudative AMD (OR = 24.45, *p* < 0.0001). Behniafard et al. reported that the GG haplotype at TNF-α (-308, -238) was seen in 92.7% of the patients with atopic dermatitis, which was significantly higher than the controls (*p* < 0.001), while a negative haplotypic association with atopic dermatitis was seen for TNF-α (-308, -238) AG and GA (*p* < 0.01) [30].

In our study, TNF-α serum levels were measured in patients with exudative AMD before the treatment of anti-VEGF and in the control groups. TNF-α protein level was lower in subjects with exudative AMD than in controls (16.182 pg/mL ± 6.094 vs. 30.652 pg/mL ± 6.322; *p* < 0.001). The expression of the TNF gene in human macrophages is strongly downregulated when they are exposed to phagocytose retinal pigment epithelial cells in vitro [31], which cellular event frequently occurs in exudative AMD.

However, our results are contradictory to previous research by scientists. Guo et al. revealed that a higher systemic level of inflammatory cytokines, including TNF-α, was associated with AMD than the control group [32]. Nagineni et al. demonstrated that TNF-α increases the secretion of vascular endothelial growth factor (VEGF) A and C by human RPE cells and choroidal fibroblasts, with VEGF being the most important factor for initiating pathological ocular neovascularization [33].

According to studies by other researchers, a decreased TNF-α protein serum level correlates with bodyweight loss. In a study of 27 obese subjects with a body mass index (BMI) of 36.3 ± 5.7, a significant decrease in TNF-α serum level was observed comparing subjects before and after weight loss treatment (7.79 pg/mL ± 4.65 vs. 6.09 pg/mL ± 2.96; *p* < 0.01) [34]. Fan et al. reported that TNF-α level was lower in chronic ketamine users than in controls (*p* < 0.05). Additionally, decreased serum TNF-α level in patients with chronic schizophrenia was correlated with an increased likelihood of having psychopathological symptoms [35].

Thus, our study demonstrated that the *TNF-α* promoter polymorphisms (rs1800630, rs1800629, and rs361525) could be potential diagnostic biomarkers of the exudative AMD. Changes in TNF-α level may also be a significant risk factor for exudative AMD.

## 5. Conclusions

*TNF-α* rs1800630 A allele was more frequent in the exudative AMD patients than in the controls. The haplotype A-A-G of the TNF-863A/C (rs1800630), TNF-308A/G (rs1800629), and TNF-238A/G (rs361525) polymorphisms were associated with the decreased odds of exudative AMD and the haplotype A-G-G with the increased odds of exudative AMD occurrence. The *TNF-α* rs1800630 A allele was statistically more frequent in females with exudative AMD than in the ophthalmologically healthy females. Additionally, the *TNF-α* rs1800630 polymorphism A allele was more frequent in females with exudative AMD than in females with primary AMD. TNF-α serum level was lower in patients with exudative AMD than in healthy controls.

## Figures and Tables

**Figure 1 life-12-00928-f001:**
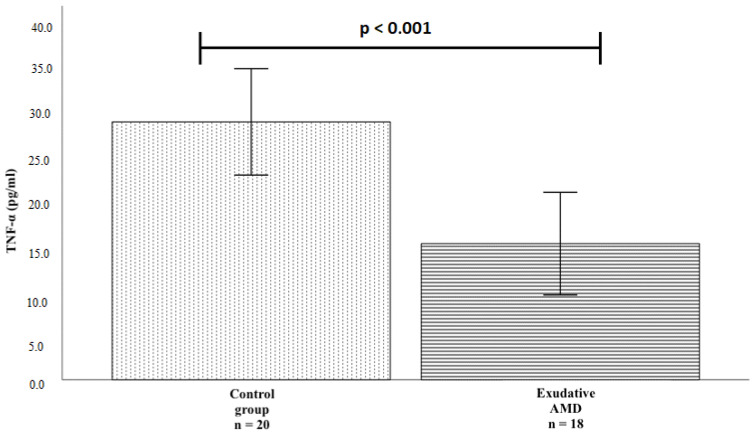
TNF-α protein serum concentrations in the exudative AMD and control groups. The bars represent the mean value of TNF-α protein concentration (mean of control group = 30.652, mean of exudative AMD = 16.182); Vertical lines represent standard deviation (SD) (control group SD = 6.322, exudative AMD SD = 6.094); *p*-value: significance level (statistically significant when *p* < 0.05); *p* < 0.001.

**Table 1 life-12-00928-t001:** Demographics.

	Control Group n = 355	Early AMD n = 330	Exudative AMD n = 393	*p*-Value
Female, n (%)	222 (62.5)	227 (68.8)	254 (64.6)	* 1–0.085 * 2–0.552
Male, n (%)	133 (37.5)	103 (31.2)	139 (35.4)	
Age, median/min./max.	72/51/94	73/42/94	77/49/95	** 1–0.266 ** 2–<0.001

* Pearson Chi-Square was used to compare two groups: 1—control group vs. early AMD, 2—control group vs. exudative AMD; ** –Whitney U test was used to compare: 1—control group vs. early AMD, 2—control group vs. exudative AMD; *p*-value: significance level (statistically significant when *p* < 0.05); min.: the lowest value; max.: the maximum value.

**Table 2 life-12-00928-t002:** Distributions of rs1800630, rs1800629, and rs361525 genotypes and alleles in patients with exudative AMD and control groups.

SNP	Genotypes/ Alleles	Group		*p*-Value
		Control n (%) (n = 355)	Exudative AMD n (%) (n = 393)	
**TNF-863A/C** **(rs1800630)**	CC	258 (72.7)	263 (66.9)	0.087
	AC	87 (24.5)	108 (27.5)	
	AA	10 (2.8)	22 (5.6)	
	C	603 (84.9)	634 (80.7)	
	A	107 (15.1)	152 (19.3)	**0.029**
**TNF-308A/G** **(rs1800629)**	GG	277 (78.0)	297 (75.6)	0.730
	AG	74 (20.8)	91 (23.2)	
	AA	4 (1.1)	5 (1.3)	
	G	628 (88.5)	685 (87.2)	
	A	82 (11.5)	101 (12.8)	0.443
**TNF-238A/G** **(rs361525)**	GG	316 (89.0)	361 (91.9)	0.310
	AG	38 (10.7)	30 (7.6)	
	AA	1 (0.3)	2 (0.5)	
	G	670 (94.4)	752 (95.7)	
	A	40 (5.6)	34 (4.3)	0.244

SNP: single-nucleotide polymorphism; AMD: age-related macular degeneration; *p*-value: significance level (statistically significant when *p* < 0.05).

**Table 3 life-12-00928-t003:** Linkage disequilibrium between studied polymorphisms in patients with early AMD.

	rs1800630 (D′; r^2^)	rs1800629 (D′; r^2^)	rs361525 (D′; r^2^)
rs1800630 (D′; r^2^)	.	0.832; 0.424	0.993; 0.009
rs1800629 (D′; r^2^)	.	.	0.996; 0.016
rs361525 (D′; r^2^)	.	.	.

D′: the deviation between the expected haplotype frequency and the observed frequency; r^2^: the square of haplotype frequency correlation coefficient.

**Table 4 life-12-00928-t004:** Linkage disequilibrium between studied polymorphisms in patients with exudative AMD.

	rs1800630 (D′; r^2^)	rs1800629 (D′; r^2^)	rs361525 (D′; r^2^)
rs1800630 (D′; r^2^)	.	0.414; 0.138	0.994; 0.011
rs1800629 (D′; r^2^)	.	.	0.995; 0.013
rs361525 (D′; r^2^)	.	.	.

D′: the deviation between the expected haplotype frequency and the observed frequency; r^2^: the square of haplotype frequency correlation coefficient.

**Table 5 life-12-00928-t005:** Haplotype association with the predisposition to exudative AMD occurrence.

TNF-863A/C (rs1800630)	TNF-308A/G (rs1800629)	TNF-238A/G (rs361525)	Frequency		OR (95% CI)	*p*-Value
			Control	Exudative AMD		
C	G	G	0.691	0.657	1.00	–
C	A	G	0.104	0.112	1.13 (0.79–1.62)	0.5
A	A	G	0.134	0.055	0.12 (0.05–1.29)	<0.0001
A	G	G	0.016	0.131	24.45 (9.39–63.63)	<0.0001
C	G	A	0.056	0.043	0.78 (0.47–1.28)	0.32
C	A	A	0	0.003	–	–

OR: odds ratio; CI: confidence interval; *p*-value: significance level (statistically significant when *p* < 0.05).

**Table 6 life-12-00928-t006:** Distributions of rs1800630, rs1800629, and rs361525 genotypes and alleles in patients with exudative AMD and control groups by gender.

SNP	Genotypes/ Alleles	Female		*p*-Value	Male		*p*-Value
		Control n (%) (n = 222)	Exudative AMD n (%) (n = 254)		Control n (%) (n = 133)	Exudative AMD n (%) (n = 139)	
**TNF-863A/C (rs1800630)**	CC	163 (73.4)	163 (64.2)	0.094	95 (71.4)	100 (71.9)	0.088
	AC	50 (22.5)	76 (29.9)		37 (27.8)	32 (23.0)	
	AA	9 (4.1)	15 (5.9)		1 (0.8)	7 (5.0)	
	C	376 (84.7)	402 (79.1)		227 (85.3)	232 (83.5)	
	A	68 (15.3)	106 (20.9)	**0.027**	39 (14.7)	46 (16.5)	0.545
**TNF-308A/G (rs1800629)**	GG	174 (78.4)	191 (75.2)	0.688	103 (77.4)	106 (76.3)	0.914
	AG	46 (20.7)	61 (24.0)		28 (21.1)	30 (21.6)	
	AA	2 (0.9)	2 (0.8)		2 (1.5)	3 (2.2)	
	G	394 (88.7)	443 (87.2)		234 (88.0)	242 (87.1)	
	A	50 (11.3)	65 (12.8)	0.469	32 (12.0)	36 (12.9)	0.746
**TNF-238A/G (rs361525)**	GG	199 (89.6)	233 (91.7)	0.460	117 (88.0)	128 (92.1)	0.115
	AG	22 (9.9)	21 (8.3)		16 (12.0)	9 (6.5)	
	AA	1 (0.5)	0 (0.0)		0 (0.0)	2 (1.4)	
	G	420 (94.6)	487 (95.9)		250 (94.0)	265 (95.3)	
	A	24 (5.4)	21 (4.1)	0.356	16 (6.0)	13 (4.7)	0.487

SNP: single-nucleotide polymorphism; AMD: age-related macular degeneration; *p*-value: significance level (statistically significant when *p* < 0.05).

**Table 7 life-12-00928-t007:** Distributions of rs1800630, rs1800629, and rs361525 genotypes and alleles in patients with early and exudative AMD by gender.

SNP	Genotypes/ Alleles	Female		*p*-Value	Male		*p*-Value
		Early AMD n (%) (n = 227)	Exudative AMD n (%) (n = 254)		Early AMD n (%) (n = 103)	Exudative AMD n (%) (n = 139)	
**TNF-863A/C (rs1800630)**	CC	169 (74.4)	163 (64.2)	0.051	66 (64.1)	100 (71.9)	0.371
	AC	49 (21.6)	76 (29.9)		32 (31.1)	32 (23.0)	
	AA	9 (4.0)	15 (5.9)		5 (4.9)	7 (5.0)	
	C	387 (85.2)	402 (79.1)		164 (79.6)	232 (83.5)	
	A	67 (14.8)	106 (20.9)	**0.014**	42 (20.4)	46 (16.5)	0.279
**TNF-308A/G (rs1800629)**	GG	185 (81.5)	191 (75.2)	0.056	82 (79.6)	106 (76.3)	0.824
	AG	37 (16.3)	61 (24.0)		19 (18.4)	30 (21.6)	
	AA	5 (2.2)	2 (0.8)		2 (1.9)	3 (2.2)	
	G	407 (89.6)	443 (87.2)		183 (88.8)	242 (87.1)	
	A	47 (10.4)	65 (12.8)	0.238	23 (11.2)	36 (12.9)	0.553
**TNF-238A/G (rs361525)**	GG	207 (91.2)	233 (91.7)	0.831	96 (93.2)	128 (92.1)	0.473
	AG	20 (8.8)	21 (8.3)		7 (6.8)	9 (6.5)	
	AA	0 (0.0)	0 (0.0)		0 (0.0)	2 (1.4)	
	G	434 (95.6)	487 (95.9)		199 (96.6)	265 (95.3)	
	A	20 (4.4)	21 (4.1)	0.835	7 (3.4)	13 (4.7)	0.485

SNP: single-nucleotide polymorphism; AMD: age-related macular degeneration; *p*-value: significance level (statistically significant when *p* < 0.05).

## Data Availability

Not applicable.

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
