# Peer review of "Associations of Tumor Necrosis Factor-Alpha Gene Polymorphisms *(TNF)-α* TNF-863A/C (rs1800630), TNF-308A/G (rs1800629), TNF-238A/G (rs361525), and TNF-Alpha Serum Concentration with Age-Related Macular Degeneration"

_life, 2022, doi:10.3390/life12070928_

Round 1

Author Response

Dear Editor and Reviewers, 

We kindly appreciate the revision of our manuscript. We have highlighted the changes we made in the manuscript by using the track changes mode in MS Word. Hope that the revised manuscript will be acceptable for publication in your journal. Enclosed please also find attached our point-by-point response to the comments raised by the reviewers (editors). 

1st reviewer 

 Major concerns: 

  1. The introduction is tediously long and not directly related to the context of the manuscript. 

The introduction was shortened. 

2. Inclusion and exclusion criteria should be provided in the Methods section, especially how did the authors define early AMD and exudative AMD. 

Information was added: 

AMD group consisted of subjects who underwent ophthalmological evaluation and were diagnosed with early or exudative AMD. Present study subjects were evaluated by slit-lamp biomicroscope. Also, all AMD patients underwent optical coherence tomography (OCT), and optical coherence tomography angiography (OCT-A) was performed to confirm the exudative AMD after the OCT examination. 

The Age-Related Eye Disease Study (AREDS) classification system was used for AMD diagnosis and classification The Age-Related Eye Disease Study system for classifying age-related macular degeneration from stereoscopic color fundus photographs: the Age-Related Eye Disease Study Report Number 6. Am J Ophthalmol 2001;132:668–81. https://doi.org/ 
10.1016/S0002-9394(01)01218-1.  

  1. Early AMD consists of a combination of multiple small drusen (protein and lipid deposit) formation between the RPE and BrM Smith W, Assink J, Klein R, Mitchell P, Klaver CCW, Klein BEK, et al. Risk factors for age-related macular degeneration: Pooled findings from three continents. Ophthalmology 2001;108:697–704. https://doi.org/10.1016/S0161-6420(00)00580-7. and several intermediate (63–124 μm in diameter) drusen, or retinal pigment epithelial abnormalities; may not cause any symptoms.  
  2. The intermediate form is described as a presence of at least one large (≥125 μm in diameter) drusen, numerous medium-sized drusen, or GA without extension to the center of the macula with mild symptoms as mild blurriness in their central vision or trouble seeing in low lighting or may not cause any symptoms. 
  3. The advanced AMD is divided into:
  • dry/atrophic AMD with the GA of the RPE;
  • neovascular or exudative AMD, which is diagnosed when choroidal neovascularization with detachments in the RPE haemorrhages and/or scars appear and cause progressive blurring or other central vision impairments Charles P. Wilkinson APSDRHSJRPW. Retina. vol. 3. 4th ed. Elsevier-Health Sciences Division; 2006. 

AMD exclusion criteria 

  1. Unrelated eye disorders, e.g., high refractive error, cloudy cornea, lens opacity (nuclear, cortical, or posterior subcapsular cataract) except minor opacities, keratitis, acute or chronic uveitis, glaucoma, or diseases of the optic nerve. 
  2. Any other inflammatory diseases. 
  3. Systemic illnesses, e.g., diabetes mellitus, malignant tumours, systemic connective tissue disorders, chronic infectious and non-infectious diseases, hypertension, coronary artery disease, stroke or conditions following organ or tissue transplantation. 
  4. Ungraded colour fundus photographs resulting from obscuring the ocular optic system or because of fundus photograph quality. 
  5. Use of antiepileptic or sedative drugs. 

  1. The discussion section should be thoroughly re-organized. It should be focused on the new knowledge added and the implication of the results, rather than a layout of the results. 

It was corrected. 

  1. The authors found a reduced serum TNF-α concentration in AMD patients. However, a lower TNF-α level seems to be a protective factor for AMD, and anti-TNF-α therapy is even a potential treatment to prevent the pathogenesis of wet AMD. What’s the reason and implication of a lower serum TNF-α level? 

It is known  that lower TNF-α serum levels is associated with an increase in visual acuity after anti-VEGF therapy. This suggests that targeting pro-inflammatory cytokines may augment treatment for neovascular AMD [Khan, A. H., Pierce, C. O., De Salvo, G., Griffiths, H., Nelson, M., Cree, A. J., Lotery, A. J. The effect of systemic levels of TNF-alpha and complement pathway activity on outcomes of VEGF inhibition in neovascular AMD. Eye,2021;1-8.]. 

Minors: 

  1. Line 81-84, please give the reference. 

The reference was added: 

Twin studies demonstrating greater concordance in monozygotic (37%) than dizygotic (19%) twins and studies showing clustering of AMD in families, a hallmark of a disease with complex inheritance, were the first to underscore the genetic basis for AMD [Hammond CJ, Webster AR, Snieder H, Bird AC, Gilbert CE, Spector TD. Genetic influence on early age-related maculopathy: a twin study. Ophthalmology. 2002 Apr;109(4):730-6.]. It has been shown that an individual with a sibling or a parent with AMD is 12-27 times more susceptible than someone from the general population to develop AMD [13]. 

  1. Line 119, the authors mentioned early AMD, should it be exudative AMD? 

Yes, it was corrected. 

  1. Did the ELISA procedure was repeated? 

No, ELISA was not repeated. 

  1. Line 200, any differences were NOT found…? 

  It was corrected.

Updated manuscript added.

 Best regards,

Greta Gedvilaite

Reviewer 2 Report

This manuscript describes the association between three SNPs of the TNF gene and the development of early or exudative stage of age-related macular degeneration (AMD). The data presented here is clinically relevant, the sample size is high, and the manuscript is well written. However, a few points should be clarified before the final publication of the manuscript.

Specific suggestions:

1.                  In the beginning of the Abstract, the objective of the study should be briefly described.

2.                  L104: “AMD” instead of ”age-related macular degeneration”

3.                  L122-123: The abbreviation of DNA should be defined at its first appearance in the text.

4.                  L146 and L148: “RT-PCR” instead of “real-time polymerase chain reaction”

5.                  2.5. The catalogue number of the applied ELISA kit should be included.

6.                  L192: “CI” instead of “confidence interval (CI)”

7.                  L200-201: The sentence should be revised for clarification.

8.                  L247-252: The sentence should be grammatically revised.

9.                  L324: The abbreviation PTB should be defined.

10.              The abbreviation SNP is not defined and not used consequently in the manuscript text.

11.              The observation that the serum TNFα levels of patients with exudative AMD is lower than of the controls might be underlined by the findings that the expression of TNF gene in human macrophages is strongly downregulated when they are exposed to phagocytose retinal pigment epithelial cells in vitro (PMID: 25450174), which cellular event frequently occurs in exudative AMD.

Author Response

2nd reviewer

Dear Editor and Reviewers, 

We kindly appreciate the revision of our manuscript. We have highlighted the changes we made in the manuscript by using the track changes mode in MS Word. Hope that the revised manuscript will be acceptable for publication in your journal. Enclosed please also find attached our point-by-point response to the comments raised by the reviewers (editors). 

 This manuscript describes the association between three SNPs of the TNF gene and the development of early or exudative stage of age-related macular degeneration (AMD). The data presented here is clinically relevant, the sample size is high, and the manuscript is well written. However, a few points should be clarified before the final publication of the manuscript. 

Specific suggestions: 

  1. In the beginning of the Abstract, the objective of the study should be briefly described. 

 It was corrected 

  1. L104: “AMD” instead of ”age-related macular degeneration” 

It was corrected 

  1. L122-123: The abbreviation of DNA should be defined at its first appearance in the text. 

It was corrected 

  1. L146 and L148: “RT-PCR” instead of “real-time polymerase chain reaction” 

It was corrected 

  1. 2.5. The catalogue number of the applied ELISA kit should be included. 

It was corrected 

  1. L192: “CI” instead of “confidence interval (CI)” 

It was corrected 

  1. L200-201: The sentence should be revised for clarification. 

It was corrected 

  1. L247-252: The sentence should be grammatically revised. 

It was corrected 

  1. L324: The abbreviation PTB should be defined. 

It was corrected 

  1. The abbreviation SNP is not defined and not used consequently in the manuscript text. 

It was corrected 

  1. The observation that the serum TNFα levels of patients with exudative AMD is lower than of the controls might be underlined by the findings that the expression of TNF gene in human macrophages is strongly downregulated when they are exposed to phagocytose retinal pigment epithelial cells in vitro (PMID: 25450174), which cellular event frequently occurs in exudative AMD. 

This information was added to the manuscript. 

Round 2

Reviewer 1 Report

I am glad  to see the revisions which made the manuscript more logistic and readable.